# Phytohormone Production by the Endophyte *Bacillus safensis* TS3 Increases Plant Yield and Alleviates Salt Stress

**DOI:** 10.3390/plants13010075

**Published:** 2023-12-26

**Authors:** Vladimir K. Chebotar, Alexander N. Zaplatkin, Elena P. Chizhevskaya, Maria S. Gancheva, Gerben P. Voshol, Natalia V. Malfanova, Maria E. Baganova, Yuriy V. Khomyakov, Veronika N. Pishchik

**Affiliations:** 1All-Russia Research Institute for Agricultural Microbiology, Podbel’skogo Shosse 3, Pushkin, 196608 St. Petersburg, Russia; pisemnet-@mail.ru (A.N.Z.); chizhevskaya@yandex.ru (E.P.C.); m.gancheva@spbu.ru (M.S.G.); mashul991@mail.ru (M.E.B.); veronika-bio@rambler.ru (V.N.P.); 2Department of Genetics and Biotechnology, Faculty of Biology, Saint Petersburg State University, 199034 St. Petersburg, Russia; 3Institute of Biology Leiden, Sylviusweg 72, 2333 BE Leiden, The Netherlands; vosgb2019@inno-mir.com (G.P.V.); natamalfanova@mail.ru (N.V.M.); 4Agrophysical Scientific Research Institute, Grazhdansky pr. 14, 195220 St. Petersburg, Russia; himlabafi@yandex.ru

**Keywords:** endophyte, *Bacillus safensis*, salt stress, phytohormones genome analysis

## Abstract

Endophytic bacteria can be used to overcome the effect of salinity stress and promote plant growth and nutrient uptake. *Bacillus safensis* colonizes a wide range of habitats due to survival in extreme environments and unique physiological characteristics, such as a high tolerance for salt, heavy metals, and ultraviolet and gamma radiations. The aim of our study was to examine the salt resistance of the endophytic strain TS3 *B. safensis* and its ability to produce phytohormones and verify its effect on plant yield in field trials and the alleviation of salt stress in pot experiments. We demonstrate that the strain TS3 is capable of producing enzymes and phytohormones such as IAA, ABA and tZ. In pot experiments with radish and oat plants in salinization, the strain TS3 contributed to the partial removal of the negative effect of salinization. The compensatory effect of the strain TS3 on radish plants during salinization was 46.7%, and for oats, it was 108%. We suppose that such a pronounced effect on the plants grown and the salt stress is connected with its ability to produce phytohormones. Genome analysis of the strain TS3 showed the presence of the necessary genes for the synthesis of compounds responsible for the alleviation of the salt stress. Strain *B. safensis* TS3 can be considered a promising candidate for developing biofertilizer to alleviate salt stress and increase plant yield.

## 1. Introduction

The world population is predicted to cross nine billion in the next few decades. Abiotic stresses such as drought, high soil salinity, heat, cold, oxidative stress and heavy metal toxicity severely hamper agriculture and are the major causes for loss in crop productivity [1,2,3]. It was postulated that salt-affected soils cover more than 800 million ha of the world’s land [4]. Salination is the result of salt accumulation over long periods of time in arid and semi-arid regions or human-induced causes, affecting about 32 million ha of farmed areas and 45 million ha of irrigated lands in the world [5,6]. Sodium chloride is the most soluble and abundant salt released [5]. Salinity is a continuous process, and its remediation is cost- and labor-intensive. It is a complex global problem [7,8,9], leading to a huge drop in soil, water, and cultivated crop quality and productivity [10,11]. Salinity stress causes various physiological changes in plants, such as interruption of membranes, nutrient imbalance, impairing the ability to detoxify reactive oxygen species (ROS), differences in the antioxidant enzymes, decreased photosynthetic activity, and a decrease in the stomatal aperture [5,12]. It was reported that, depending on the plant species and the growth stage, it may affect stresses, with a sodium chloride content of more than 200 mM sometimes leading to its subsequent mortality [13,14,15].

It is believed that endophytes evolved as an intermediate group between saprophytic bacteria and plant pathogens, and any of the 300,000 plant species existing in the world can be the host of one or more endophyte species. [16]. The high halotolerance among the endophytes themselves for plants growing in a salty environment suggests that the plant can more easily adapt to stress along with different bacterial populations [17,18]. Therefore, endophytic bacteria can be used as biofertilizers, alleviating salt stresses due to their direct interaction with the host plant, leading to less competition with other microbes [19,20,21,22].

Stress responses by plants tend to be mediated by the production of phytohormones [23]. The most notable plant growth-promoting hormones that can be synthesized by bacteria include indole-3-acetic acid, zeatin, abscisic acid, cytokinin, gibberellic acid, and ethylene [24].

Auxin indole-3-acetic acid (IAA) is a significant phytohormone produced by a number of endophytic bacterial species colonizing halophytic plants, and IAA acts by increasing plant seed germination, root proliferation, and cell permeability to water and by decreasing cell wall pressure [25]. In addition to promoting growth directly through the production of IAA, microorganisms can also promote plant growth by improving the plant resistance to abiotic stresses. For instance, Soltani et al. [26] showed that strains of halotolerant endophytic bacteria could produce IAA to increase the salt resistance of corn and wheat seeds during germination. Soybeans inoculated with the IAA-producing strain YNA 40 *Acinetobacter pitti* showed a significant improvement in plant growth with salt [27]. Some IAA-producing bacterial strains isolated from alkali-salinated soils can increase the salt resistance of maize and wheat seeds during germination. In addition, it was shown that the production of IAA by bacteria contributed to the growth and salt resistance of perennial ryegrass, as well as upland cotton and pak choi plants [28,29,30,31].

In addition to IAA, endophytic bacteria can produce other hormones, such as abscisic acid (ABA) [24]. ABA causes stomata closure in plant leaves, thereby reducing water loss [32]. Therefore, plants are able to increase the biosynthesis of abscisic acid (ABA) during salinization [23,33]. Cytokinin (cis-zeatin cytokinin, riboside type zeatin, isopentyladenine, and isopentenyladenosine) synthesis was detected in endophytes isolated from cucumber (*Cucumis sativus* L.). These bacteria belonging to the genera *Pseudomonas*, *Sphingomonas*, *Stenotrophomonas*, and *Arthrobacter* were able to produce several cytokinins, the content of which exceeded 30 pmol/mL [34].

The bacterium species *Bacillus safensis* is a member of the ubiquitous, Gram-positive, endospore-forming genus *Bacillus*. The first type strain (FO-36b) was originally isolated in the spacecraft assembly facility (SAF) at the Jet Propulsion Laboratory (Pasadena, CA, USA), from which it derived its species name “safensis” [35]. Members of this species colonize a wide range of habitats, many of which are stringent for the survival of microorganisms. Its survival in extreme environments relies on its unique physiological characteristics, such as a high tolerance for salt, heavy metals, and ultraviolet and gamma radiations and resistance to sterilizing agents, including ethanol and hydrogen peroxide [36,37].

Along with the ability to survive under conditions of abiotic stress, *Bacillus safensis* strains may have plant growth-promoting properties. For example, it was shown in pot experiments with wheat plants (*Triticum aestivum* L.) under the salinization of 200 mM NaCl that application of *Bacillus safensis* W10 to the rhizosphere of wheat plants increased the shoot dry weight by 2.09–2.81 times and the root dry weight by 2.16–2.83 times compared with uninoculated control [38].

Recently, we isolated the endophyte *B. safensis* TS3 from surface-sterilized stems of greenhouse tomato (*Lycopersicon esculentum* Mill.) plants cv. Carmello grown in salty soils [39]. Analysis of the TS3 genome sequence showed that a large part of the genome (>6%) is connected with the production of various plant growth-promoting and stress-alleviating metabolites such as IAA, volatiles, organic acids, vitamins, compatible solutes, exopolysaccharides, and antioxidant enzymes.

The aim of our study was to study the properties of the endophytic strain TS3 *Bacillus safensis*, such as salt resistance, the ability to produce phytohormones, the type of plant colonization, and the ability to stimulate the growth of various plants and reduce salt stress, as well as analysis of the genome of the strain TS3 in order to find the genes responsible for its properties.

## 2. Results

### 2.1. Production of Enzymes and Phytohormones by the Strain TS3

It was found that the strain TS3 of *Bacillus safensis* was capable of producing proteases, lipases, beta-glucanases, and cellulases. The halo zones on the selected nutrient media around the colonies of TS3 are presented in Table 1.

The production of phytohormones by the strain TS3 was estimated by high-performance liquid chromatography. The concentrations of indole 3-acetic acid (IAA), trans-zeatin (tZ), and abscisic acid (ABA) in the bacterial suspension of *B. safensis* TS3 are presented in Table 2. It was demonstrated that the strain TS3 produced considerable amounts of phytohormones, namely 366.1 µg/L of IAA, 0.646 µg/L of ABA, and 0.822 µg/L of tZ.

### 2.2. Colonization Ability of Bacillus safensis TS3

The colonization ability of the strain TS3 of *B. safensis* was studied in model experiments with rapeseed plants. Surface sterilized seeds were treated with a spore suspension of the strain TS3, and then the number of bacteria on the treated seeds was analyzed. The aim of the experiment was to study how the strain TS3 was able to colonize the seeds of the plant and proliferate on the seeds’ surface. We found that strain TS3 dominated in the bacterial microbiome of the treated seeds (Figure 1). The number of bacilli including both the vegetative cells and spores in the variant with TS3 treatment was 3.6 times higher, comparing with the number of the spores alone which was 2.6 times higher than that in the control (water treatment). These data mean that some of the TS3 spores used for seed treatment were able to germinate and attach to the rapeseed seeds. (This was indicated by a larger number for the cells and spores compared with only spores).

Further analysis of the rapeseed roots showed that the TS3 strain was able to successfully colonize the roots and endosphere of plants. This was indicated by an increase in the number of root- and endosphere-cultivated bacteria in plants treated with the TS3 strain compared with the untreated control (Figure 2).

It is interesting to note that strain TS3 more effectively colonized the endopshere of oilseed rape roots than the roots’ surface compared with the control plants. Thus, the amount of TS3 in the endopshere was 1.52 times greater, while on the roots’ surface, it was only 1.15 times greater compared with the untreated control plants (Figure 2).

This effect was also demonstrated in the CLSM pictures (Figure 3a,b). *B. safensis* TS3 demonstrated very good root colonization, especially at the mature root hair zone, as well as the endophytic mode of colonization (arrows in Figure 3).

In root colonization, the TS3 strain demonstrated various types of colonies (Figure 4a,b). It formed biofilms and densely packed colonies (Figure 4a) as well as loose colonies mainly on the root hair (Figure 4b).

### 2.3. The Effect of Different Concentrations of NaCl on the Growth of TS3

We analyzed the ability of strain TS3 to grow with 1%, 5%, 10%, 15%, and 20% NaCl. The number of cells grown in a liquid LB medium with the addition of various concentrations of NaCl is shown in Table 3. It is shown that TS3 was able to grow on the LB medium with 1–15% NaCl but did not grow at 20% NaCl.

### 2.4. The Growth-Stimulating Activity of Strain TS3 under the Condition of an Increased NaCl Concentration Using Gnotobiotic Systems

Experiments to study the effect of sodium chloride on the development of the seedlings of agricultural plants (oats, wheat, and radish) revealed the concentrations of NaCl at which plant development was not completely suppressed, but their noticeable inhibitory effect was observed (Table 4).

In the experiments with salinization, it was shown that the introduction of 0.5% NaCl did not significantly affect the development of the seedlings for all tested crops. However, at 1% NaCl, there was already a significant suppression of plant development by more than 50% according to such indicators as the “average shoot length” and “average root length”. At NaCl concentrations of 1.5 and 2%, inhibition became even more significant, and plant development almost completely stopped. The concentration of 2.5% completely suppressed the germination of radishes, and 5% completely suppressed the germination of cereals. For further experiments with artificial salinization, a concentration of sodium chloride equal to 1% was chosen. (Soils containing more than 0.25% water-soluble salts from the total weight of dry soil were considered saline.)

We studied the growth-stimulating activity of strain TS3 under the conditions of increased NaCl content in winter wheat plants using a gnotobiotic system [40]. A variant without NaCl was considered the negative control, while a variant with a content of 1% NaCl was considered the positive control. To identify the dependence of the growth-stimulating effect on the number of bacteria, the seeds were treated with different concentrations of the TS3 strain: 10^5^, 10^6^, and 10^7^ CFU/mL. Under salinization conditions, the growth-stimulating activity of strain TS3 showed an increase in the number of cells from 10^5^ to 10^7^ CFU/mL in a bacterial suspension used for seed treatment of winter wheat plants (Table 5 and Figure 5).

The shoot length in the variant with NaCl (positive control) decreased by 84.3% compared with the variant without NaCl (negative control). However, seed treatment of the winter wheat plants with strain TS3 helped the plants overcome salinity stress by 226.9–284.6% compared with the positive control. The compensatory effect of strain TS3 for the treated plants ranged from 35.5% to 44.5%, depending on the cell number in the bacterial suspension used for seed treatment of the winter wheat plants. The same effect of strain TS3 was observed for the root length. The compensatory effect of strain TS3 varied from 27.9% to 29.1%, depending on the cell number in the bacterial suspension.

### 2.5. Pot Experiments with Strain TS3 on Salinated Soil

We studied the effect of inoculation with strain TS3 of *Bacillus safensis* on oats cv. Borrus and radish cv. Duro under salinization conditions. The results of the experiments show that salinization significantly reduced the biomass of the radish fruits. Thus, in the positive control with the addition of sodium chloride, the weight of the root crops decreased by more than three times to 28.6% of the weight of the negative control without salinization. However, inoculation with strain TS3 contributed to the partial removal of the negative effect of salinization (Figure 6). Based on the data obtained, a diagram of the compensatory capabilities of the TS3 strain was compiled, which allowed us to assess how much endophytic bacteria can relieve plant stress from salinization (Figure 6).

Thus, the compensatory effect on radish plants during salinization under inoculation with the TS3 strain was 46.7%. This was 18.1% more than the value in the positive control. It should be noted that the radish root crops in the inoculation variant were aligned in shape and size and had a more marketable appearance than the root crops in the negative control without sodium chloride. This indicates that salinated soils can also produce commercial agricultural products that do not differ from products obtained from unsalinated soils.

An even more pronounced compensatory effect of the TS3 strain was observed in oats cv. Borrus (Figure 7a). Thus, inoculation with the TS3 strain gave an increase of 8% relative to the negative control without the addition of sodium chloride, and the increase from inoculation with this strain relative to the positive control (salinization) was 36%. In addition, the inoculated plants were ahead of the positive control plants in terms of development, as the treated oat plants were spiked, while the uninoculated plants grown under salinization conditions were at the stage of incomplete ear sweeping (Figure 7b).

### 2.6. Field Tests with Potato and Cabbage Plants

Field tests of strain TS3 of *Bacillus safensis* with potato and cabbage plants demonstrated a pronounced effect on plant yield (Figure 8) in the variants with maximal doses of bacterial suspension. Thus, preplanting treatment of potato tubers with a dose of 1.0 L/t and double spraying of plants with a dose of 3.0 L/ha resulted in a yield increase of 8.4% or 3.4 t/ha. Spraying the cabbage plants before planting seedlings with a dose of 1.0 L/ha and double spraying cabbage plants with a dose of 4.5 L/ha resulted in a yield increase of 29.4% or 14.5 t/ha.

### 2.7. Genome Analysis

Recently, the genome of the strain TS3 *Bacillus safensis* was sequenced [39]. It has been shown that more than 6% of the TS3 genome is associated with the production of various metabolites capable of stimulating plant growth and relieving abiotic stresses. Such metabolites include IAA, volatile substances, organic acids, vitamins, compatible solutes, exopolysaccharides, antioxidant enzymes, and metal reductases. The genome analysis of TS3 revealed the presence of three complete biosynthetic pathways for IAA synthesis, namely the IPA, IAN, and TAM pathways (Figure 9). The IPA pathway is the most common pathway found in various plant growth-promoting bacteria and relies on the conversion of indole-3-pyruvate (IPA) into indole-3-acetaldehyde (by BISP_00666) and consequently to IAA (by BISP_00055). Aside from the IPA pathway, we found genes encoding the IAN pathway. The IAN pathway is found in both bacteria and plants and consists of an IAA acetyltransferase (BISP_03005) and a gene encoding a putative nitrilase (BISP_03443). The final pathway (TAM) involves the conversion of tryptophan to tryptamine (by BISP_00666) and a subsequent conversion to indole-3-acetaldehyde (by BISP_00055).

The genome of TS3 also contains the necessary genes for the synthesis of proline *pro*A and *pro*B (BISP_02116 and BISP_02117, respectively). Compatible solutes are either synthesized de novo (for example, proline) or taken up from the surrounding environment. Since synthesis of these compounds is an energy-intensive process, and under stress conditions, energy conservation is better for survival, the uptake of compatible solutes is preferred. For this purpose, TS3 has two different glycine betain transport systems in its genome (BISP_02550-BISP_02552 and BISP_01659-BISP_01661).

The genome of TS3 contains one gene cluster involved in EPS synthesis (BISP_01698- BISP_01702), which is important for attachment to surfaces and forming biofilms to provide physical protection from drying out.

To neutralize reactive oxygen species, TS3 is able to synthesize a number of antioxidant enzymes. These include a superoxide dismutase (BISP_00063), catalase (BISP_00541, BISP_01431, BISP_03169, and BISP_03457), and glutathione reductase (BISP_00121).

## 3. Discussion

Considerable attention has been dedicated recently to the screening of bacterial endophytes as natural biofertilizers and plant protectors [42]. Halophytic plants are the best model for the isolation of bacterial endophytes helping with the elimination of excess salt concentrations [43,44,45,46,47]. Many bacteria of the genus *Bacillus* are able to stimulate plant growth and provide protection under biotic and abiotic stresses [48,49,50,51,52].

Bacteria from the genera *Bacillus* and *Pseudomonas* are the most well-known candidates for the development of bioinoculants to combat biotic and abiotic stresses as well as stimulate plant growth [53]. However, biofertilizers containing bacteria of the genus *Bacillus* are more resistant to environmental stresses due to their spore formation ability compared with non-spore-forming bacteria of the genus *Pseudomonas*. Currently, most commercial biofertilizers contain different strains of bacilli. Among them, the most famous are Alinit (*Bacillus subtilis*), Kodiak (*Bacillus subtilis* GB03), Quantum-400 (*B. subtilis* GB03), Rhizovital (*B. amyloliquefaciens* FZB42), Serenade (*Bacillus subtilis* QST 713), and YIB (*Bacillus* spp.). However, most of them are aimed at the biocontrol of phytopathogenic microorganisms and stimulation of plant growth. Specialized biofertilizers that alleviate salt stress have not yet been developed. Therefore, the development of such biofertilizers seems to be a rather promising scientific direction.

The genus *Bacillus* Cohn (1872) (of the family Bacillacea) is a diverse group (heterogeneous taxon) whose representatives are ubiquitous in nature, and at present, this genus includes a significant number of species [54]. Bacteria of the genus *Bacillus* are usually referred to as typical soil bacteria, but numerous studies proved their ubiquitous distribution. They are found in soil, water, and on the surface and in the internal tissues of plants. Along with *B. subtilis*, other related species are also described (*B. amyloliquefaciens*, *B. siamensis*, *B. velezensis*, *B. atrophaeus*, *B. safensis*, *B. inaquosorum*, *B. licheniformis*, *B. mojavensis*, *B. paralicheniformis*, *B. sonorensis*, *B. spizizenii*, *B. stercoris*, *B. tequilensis*, *B. vallismortis*, etc.) [55].

Strains of *Bacillus safensis* have been isolated from a wide range of environments, including surface soil [56] and plant microbiomes [57]. Some strains of *B. safensis* have demonstrated antifungal activity [58] and the promotion of plant growth [59]. We isolated the endophyte *B. safensis* TS3 from the stems of tomatoes grown in salty soils. TS3 was capable of producing proteases, lipases, beta-glucanases, and cellulases as well as a range of phytohormones such as IAA, ABA, and tZ. Our data coincide with the results of other researchers. Thus, the isolation of salt-tolerant, plant-growth-promoting endophytic *B. safensis* ZY16 from the root of a halophyte *Chloris virgata* Sw., which was able to produce 31.4 mg/L IAA, was reported [60]. The endophytic bacteria *Bacillus safensis* RS95 isolated from the stem of lowland rice plants was able to produce 65.6 µg/mL IAA [61].

Many studies have been published on the effect of auxins on plant resistance to abiotic stresses and, in particular, salt stress [62]. Thus, rice plants showed a significant decrease in IAA content due to the effect of salt stress. According to other researchers, such variation in the content of IAA can induce growth modulation due to an increase in other phytohormones, such as ABA [63]. The role of auxin in modulating seed germination at a high salinity due to the incorporation of auxin signaling in the transmission of salt stress signals during seed germination has been shown [64].

Cytokinins are another important group of plant hormones involved in maintaining cellular proliferation and differentiation and the prevention of senescence [65]. However, under stress conditions, particularly water stress, an increased concentration of cytokinin in the xylem sap was observed [66]. Zhang et al. [67] demonstrated that cytokinin-overexpressing transgenic cassava exhibited greater tolerance to drought in comparison with wild-type plants.

Abscisic acid (ABA) is also known to have an important role in plants by improving stress responses and adaptation. There have been many reports describing the role of ABA in integrating signaling during stress exposure with subsequent control of the downstream responses [68]. It has been shown that inoculation of lettuce with the strain *B. subtilis* IB-22, which is capable of producing various cytokinins, led to the accumulation of cytokinins in inoculated plants and increases in the shoot and root weights of approximately 30% [69].

Thus, the production of such important phytohormones as IAA, ABA, and tZ by the TS3 strain can significantly affect the ability of the host plant to overcome abiotic stresses, including salinization. In addition, we have shown that the TS3 strain was able to effectively colonize the endosphere of plants and was a good root colonizer. These traits are quite important for the delivery of produced phytohormones to the host plants.

We demonstrated that our TS3 strain was able to grow on the nutrient medium with 15% NaCl and stimulate the growth of different plants (winter wheat, oats, and radish) under salinization conditions. For instance, the greatest effect of inoculation of winter wheat with the various concentrations of the TS3 strain in salinated sand was observed when the number of cells in the bacterial suspension used for seed treatment was increased from 10^5^ to 10^7^ CFU/mL. The compensatory effect of the TS3 strain from the salinity stress for the treated plants ranged from 35.5% to 44.5% for the shoot length and varied from 27.9% to 29.1% for the root length, depending on the cell number in the bacterial suspension used for seed treatment of the winter wheat plants. The plant growth-promoting effect of the TS3 strain of *B. safensis* with various doses of a bacterial suspension was confirmed in the field tests with potato and cabbage plants, resulting in yield increases of 4.4–8.4% and 16.6–29.4%, respectively.

Our results are consistent with the data of other researchers. Thus, it was shown that the strain ZY16 *B. safensis* demonstrated high salt resistance (up to 16% NaCl) in the nutrient medium [60]. It was reported that endophytic bacteria isolated from the internal tissues of the root, stem, and seed of peanut plants, grown in farmers’ fields affected by salinity, can withstand NaCl concentrations of up to 12.5% [70]. Therefore, high resistance to salinization is characteristic of many strains of *B. safensis*. This was confirmed in pot experiments with wheat plants (*Triticum aestivum* L.) under salinization of 200 mM NaCl when the application of *Bacillus safensis* W10 to the rhizosphere of wheat plants increased the shoot dry weight by 2.09–2.81 times and the root dry weight by 2.16–2.83 times compared with an uninoculated control [38]. These results are consistent with the data from our pot experiments with radish and oats plants under salinization.

We demonstrated that our TS3 strain contributed to the partial removal of the negative effect of salinization. The compensatory effect of the TS3 strain on radish plants during salinization was 46.7%, and on oats, it was 108%. We suppose that such a pronounced effect on the plants grown in salt stress can be connected to its ability to produce phytohormones such as IAA, ABA, and tZ. Many studies have reported the positive effects of bacteria producing IAA on plant growth stimulation under abiotic stress conditions. For example, it was demonstrated that the endophytic strain *Enterobacter* sp., which was isolated from the halophytic plant *Psoralea corylifolia* L., enhanced the salt tolerance of the non-host plant *Triticum aestivum* [71]. Treatment of durum wheat seeds with the endophytic strain Pa *Pantoea agglomerans* showed a significant positive effect on the shoot length in both the absence of salt stress and with 100 mM of salt [72]. The halotolerant bacterium *Bacillus licheniformis* HSW-16 was able to mitigate salt stress-induced damage and stimulate the growth of wheat through the production of IAA under saline soil conditions [73]. Several salt-tolerant endophytes synthesizing IAA in a culture medium, such as *Serratia plymuthica* RR-2-5-10 and *Stenotrophomonas rhizophila* e-p10, improved the cucumber biomass and yield under salt stress in greenhouse conditions by 9–24% [74].

The application of salt-tolerant endophytic bacteria able to produce IAA and associated with the halophyte New Zealand spinach (*Tetragonia tetragonioides* (Pall.) Kuntze) under saline conditions (200 mM NaCl) significantly increased the root and shoot dry weight by 27% and 35%, respectively, compared with uninoculated plants exposed to salt stress [75]. The bacterial endophyte strain EGY05 *Bacillus sonorensis*, isolated from the wild plant *Thymus vulgaris* (North Sinai, Egypt), produced auxin and increased the root fresh weight by 27.6, 19.2, 21.9, and 40.2% under 50, 100, 150, and 200 mM NaCl treatments, respectively [76].

Also, many researchers have reported stimulation of plant growth by cytokinin-producing bacteria. For instance, cytokinin-producing root-associated bacteria from the genera *Arthrobacter*, *Bacillus*, *Azospirillum*, and *Pseudomonas* increased the soybean shoot and root biomass as well as the proline content in plant tissue under salt stress [77].

A genome comparison of the coding sequences (CDSs) of TS3 versus six closely related reference *B. safensis* genomes allowed us to identify 115 proteins not present in the other six examined genomes. The genome analysis of TS3 revealed the presence of three complete pathways for IAA synthesis, namely the IPA, IAN, and TAM pathways, the content of necessary genes for the synthesis of proline (*pro*A and *pro*B), two different glycine betain transport systems, and a number of antioxidant enzymes, such as superoxide dismutase, catalase, and glutathione reductase. In addition, plant-associated bacteria secrete a mixture of volatile compounds that can enhance plant growth. The bacterial volatiles with a confirmed plant growth-promoting ability include 3-hydroxy-2-butanone (acetoin) and 2,3-butanediol [42]. Genes encoding acetolactate synthase (*als*S)- and acetolactate decarboxylase (*als*D)-catalyzing synthesis of acetoin that could be converted to 2,3-butanediol were detected in the genome of TS3.

Salt-tolerant microbes secrete polysaccharides to adhere to surfaces and for biofilm formation, which provides physical protection against desiccation. Among all the bacterial polysaccharides produced, exopolysaccharides (EPSs) have been identified as the most important ones for biofilm formation. The formation of EPSs not only protects the beneficial microbes but also helps form a sheath around the roots (rhizosheath), thereby protecting plants against salinity [78]. We demonstrated that the genome of TS3 contains a gene cluster involved in EPS synthesis.

In response to different stresses, plants accumulate large quantities of different types of compatible solutes [79]. Compatible solutes are low molecular weight, highly soluble organic compounds that are usually nontoxic at high cellular concentrations. These solutes provide protection to plants from stress by contributing to cellular osmotic adjustment, the protection of membrane integrity, and enzyme and protein stabilization [80,81]. These include proline, sucrose, polyols, trehalose, and quaternary ammonium compounds (QACs) such as glycine betaine, alinine betaine, proline betaine, and pipecolate betaine [82,83]. When faced with increases in the external osmolarity, many microorganisms amass compatible solutes to counteract water efflux. They thereby adjust the turgor to physiologically appropriate values and promote cell growth under otherwise osmotically unfavorable circumstances [84]. L-proline is one of the well-known representatives of this class of compounds [85,86]. Therefore, the possibility of biosynthesis of a significant amount of L-proline may be one of the mechanisms of resistance of the TS3 strain to high concentrations of NaCl.

Another coping mechanism that might be used by TS3 against salinity stress is the production of antioxidant enzymes. During salinity stress, a partial reduction in oxygen leads to the production of reactive oxygen species (ROS). ROS cause oxidative damage to proteins, lipids, and nucleic acid, thereby disturbing cell homeostasis. To neutralize these ROS, TS3 is able to synthesize a number of antioxidant enzymes. These include the superoxide dismutase, catalase, and glutathione reductase.

Thus, strain TS3 of *B. safensis* has many mechanisms both for its own survival at elevated NaCl concentrations and for supporting the host plant under salt stress. In this regard, strain TS3 can be considered a promising candidate for the development of a biofertilizer to reduce salt stress and increase plant yields.

We suggest that the production of phytohormones such as IAA, ABA, and tZ by the strain TS3 of *B. safensis* plays a key role in the ability to reduce salt stress and increase plant productivity. At the same time, genome analysis of the strain TS3 of *B. safensis* showed the presence of the necessary genes for the synthesis of proline, two different glycine betain transport systems, and a number of antioxidant enzymes, such as superoxide dismutase, catalase, and glutathione reductase. Therefore, the production of these compounds by the strain TS3 of *B. safensis* and its protective ability will still need to be demonstrated.

## 4. Materials and Methods

### 4.1. Estimation of Cellulase, Amylase, Protease, and Lipase Activity

The estimation of cellulase, amylase, protease, and lipase activities was performed as described in [47].

### 4.2. Analysis of Bacterial Phytohormones

*Bacillus safensis* TS3 was cultured in liquid potato-dextrose broth (PDB, Sigma, St. Louis, MO, USA) for 4 days at 28 °C in 500 mL flasks placed in a rotary shaker (180 rpm). The concentrations of indole 3-acetic acid (IAA), trans-Zeatin (tZ), and abscisic acid (ABA) in the bacterial suspension were determined using high-performance liquid chromatography (Agilent 1200, Santa Clara, CA, USA) with a mass-selective detector Varian 920-LC system as described in [49].

### 4.3. Determination of the Colonization Ability of Bacillus safensis TS3

Oilseed rape (*Brassica napus* L., cv.Avatar, NPZ Norddeutsche Pflanzenzucht Hans-Georg Lembke KG, Holtsee, Germany) was used to analyze plant colonization. Before bacterial inoculation, oilseed rape seeds were subjected to surface sterilization for 5 min in a 2% NaOCl solution and then washed six times with sterile distilled water. For seed treatment (bio-priming), *B. safensis* TS3 was grown on the following, expressed as NA, g/L: beef extract, 3.0; peptone, 5.0; and agar, 15.0 (Nutrient Agar, Sigma Aldrich, St. Louis, MO, USA). A strain was grown for 72 h, and the cells were scraped from the plates and suspended in sterile distilled water. The live cell concentration was measured by serial dilutions of the bacterial suspension followed by plating. To isolate the spores, a bacterial suspension of strain TS3 of *B. safensis* was heated to 90 °C for 30 min and then plated out on the NA. Bio-priming of the seeds was performed as described in [87]. Briefly, the seeds were immersed in the cell suspension for 4 h at 20 °C under agitation, and seeds incubated with sterile distilled water for 4 h served as a control. The infiltrated seeds were dried for 1 h at 20 °C. For determination of the cell number on the seeds, 20 seeds were transferred into 2 mL of a sterile 0.85% NaCl solution. The experiment was performed in four replications. The treated seeds were ground using an autoclaved mortar and pestle. One part of the suspensions was serially diluted and plated onto the NA (two replicates per dilution), and the other part was heated to 90 °C for 30 min before plating. The plates were incubated for 24–48 h at 30 °C, and the colony-forming units (CFUs) were counted in order to estimate the logarithmic means of the CFUs (log10 CFU).

The treated seeds were placed in germination pouches (Mega International, Minneapolis, MN, USA). Seven treated seeds with TS3 and seven untreated seeds (control) were aseptically placed into one germination pouch which had been filled with 20 mL of sterile distilled water. The prepared pouches were placed in sterile plastic boxes for 14 days at 23 ± 2 °C under artificial lighting (with a 16 h light period). After 14 days of the experiment, the roots were separated from the plants and weighed. The cell number of the TS3 strain on the oilseed rape roots was determined in eight replications, using 14 seedlings per replication. The roots of the 14 seedlings were placed in a sterile mortar, and 2 mL of NaCl solution (0.85%) was added and thoroughly homogenized with a pestle. Serial dilutions were prepared from the obtained homogenates, which were sown on the NA as described earlier. Petri dishes were incubated for 24–48 h at 30 °C, and the CFU per gram of raw roots was calculated. To count the number of endophytic bacteria, the cut oilseed rape roots were placed in ethanol for 2 min and dried on sterile filter paper, and their CFUs were determined from serial dilutions as described above.

### 4.4. Root Colonization Study by Using Fluorescence In Situ Hybridization and Confocal Laser Scanning Microscopy

Oilseed rape (*Brassica napus* L.) root colonization by *Bacillus safensis* TS3 was visualized using fluorescence in situ hybridization (FISH) and confocal laser scanning micros-copy (CLSM). The roots of the TS3-primed seedlings (14 d) were fixated with paraformaldehyde (PFA). Fluorescence in situ hybridization was carried out following the protocol from [88], using the fluorescence in situ hybridization probes EUB338 I-III (Cy3-labeled) mix for universal bacterial staining and the LGC354 A-C (Cy5-labeled) mix for firmicute staining. Confocal laser scanning microscopy was performed on a Leica TCS SPE DM5500Q microscope (Leica Microsystems, Wetzlar, Germany). The root samples were examined at three different regions: mature root with root hairs, the elongation zone, and the root tip. Then, the 3D pictures were edited using Imaris 7.30 (Bitplane, Zürich, Switzerland), and 3D videos of the colonized root surfaces were taken.

### 4.5. The Effect of Different Concentrations of NaCl on the Growth and Cell Number of TS3

The effect of NaCl on the growth of bacteria was studied on a Luria–Bertani (Thermo Fisher Scientific, Waltham, MA, USA) medium with the addition of various concentrations of NaCl (1%, 5%, 10%, 15%, and 20%) [47]. The Petri dishes were incubated at a temperature of 28 °C. The growth of bacteria was noted visually after 72 h. The effect of NaCl on the cell number of TS3 was studied in the liquid LB medium supplemented with various concentrations of NaCl (1%, 5%, 10%, 15%, and 20%). The TS3 was cultured for 3 days at 28 °C in 500 mL flasks placed in a rotary shaker (180 rpm). Then, aliquots (100 µL) of the serial dilutions (10^−1^–10^−5^) were inoculated onto LB plates. The plates were maintained at 28 °C for 3 days, and the cell number was calculated.

### 4.6. The Growth-Stimulating Activity of the TS3 Strain under the Condition of Increased NaCl Concentrations Using Gnotobiotic Systems

For the experiments with artificial salinization, a concentration of sodium chloride equal to 1% was chosen. (Soils containing more than 0.25% water-soluble salts from the total weight of dry soil were considered saline).

To create a gnotobiotic system, test tubes with sterile quartz sand were used, into which 10 g/kg (1%) of NaCl was introduced. A general view of the gnotobiotic systems model is shown in Figure 10.

For the winter wheat, the “Batko” variety of P.P. Lukyanenko National Grain Center selection was used for the experiments. The experiment with salt included a negative control without the introduction of NaCl, a positive control with the introduction of 10 g/kg of NaCl, and a variant with the TS3 strain. The experiment was performed in three replications.

Each tube was filled with 34 g of quartz sand, as well as 10 mL of a solution containing 10 g/kg of NaCl and 15% nutrient solution PNS [89]. Then, the test tubes were sterilized in an autoclave for an hour at 131 °C. The seeds were sterilized with solutions of ethanol and sodium hypochlorite. The seeds were kept in 70% ethanol solution for 2 min and washed in sterile water. Then, the seeds were placed in a 30% solution of sodium hypochlorite and placed on a shaker for 20 min. Then, they were washed in sterile water, and the procedure was repeated. The seeds, thoroughly washed in sterile water, were placed into the sterile Petri dishes on filter paper moistened with sterile water (5 mL per dish) and germinated at a temperature of 28° C for 24 h.

Inoculation of the seeds was carried out with a night (20 h) culture of the strain TS3 grown on a liquid LB (Thermo Fisher Scientific, MA, USA) medium. The sterile wheat seedlings were inoculated with a bacterial suspension with a cell concentration of 10^5^, 10^6^, or 10^7^ CFU/mL for 30 min. The inoculated seeds were planted in sand with 2 plants per test tube (6 plants in the variant) and grown for 12 days. Then, the plants were extracted and accounted for, and the lengths of the shoots and roots were measured.

### 4.7. Pot Experiments with the Strain TS3 in Salinated Soil

For the pot experiments in salinated soil, six pots with oats cv. Borrus and six pots with radish cv. Duro were used. For the negative control without the introduction of sodium chloride, a soil with a background salt content was used. In the other variants, imitation of salinization was achieved by adding sodium chloride. For an experiment to study the stimulation of plant growth in conditions of artificial salinization, the following soil composition was used: sieved sod-podzolic soil (50%), the commercial ready-made peat substrate Terra Vita (produced by LLC Fart, Saint-Petersburg, Russia) (30%), and quartz sand (20%). The pots (volume of 5 L) for the experiments with sodium chloride contained 2.9 kg of artificial soil. Sodium chloride was added at an amount of 7.5 g per pot, which in terms of the total weight of the soil was 0.258%. (Soils containing more than 0.25% water-soluble salts from the total weight of dry soil were considered saline.) NaCl was introduced into the soil with irrigation water. The sots simulating salinization were given 1% sodium chloride solution, with 750 mL of solution per pot. The non-sterile plant seeds were previously germinated at 28 °C for a day for radishes and for two days for the oats until 2–5 mm roots appeared. Inoculation of the seedlings was carried out with a night (20 h) culture of the TS3 strain grown on a liquid LB medium at a rate of 1 mL per seedling with a cell number of 10^7^ CFU/mL, after which they were planted into the soil.

In the experiments with salinization, five sprouted oat seeds were planted in one pot, and three radish seedlings were planted in one pot. Watering was carried out with water without salt as the soil dried at a volume of 400 mL per pot. Radish plants were grown for 40 days, and oats were grown for 56 days. After harvesting the radish fruits and oats, the biomass and grain were measured by weighing.

### 4.8. Field Tests with Potato and Cabbage Plants

Field tests of strain TS3 of *B. safensis* with potato and cabbage plants were carried out in the Moscow region by the All-Russian Research Institute of Potato Farming, named after A.G. Lorkh, and the All-Russian Research Institute of Vegetable Breeding and Seed Production.

Bacteria for the field tests were cultured in liquid potato-dextrose broth (PDB, Sigma, USA) for 2 days at 28 °C on a rotary shaker with 200 rpm to obtain a final concentration for the bacterial suspension of 1.5 × 10^8^ cfu/mL. Field tests with potato were carried out on cv. Sante (Agrico Company, Emmeloord, The Netherlands). The aim of the field tests was to determine which amount of the TS3 bacterial suspension (1.5 × 10^8^ cfu/mL) had the most pronounced effect on potato yields. The experimental design was as follows:Control: N100P110K130;N100P110K130 + preplanting treatment of tubers, with consumption of the bacterial suspension (1.0 L/t for consumption of the working solution and 10.0 L/t for spraying of the plants in the vegetative growth stage first and in the flowering stage second); consumption of bacterial suspension = 1.0 L/ha; consumption of working solution = 300.0 L/ha;N100P110K130 + preplanting treatment of tubers, with consumption of the bacterial suspension (1.0 L/t for consumption of the working solution and 10.0 L/t for spraying of the plants in the vegetative growth stage first and in the flowering stage second); consumption of bacterial suspension = 2.0 L/ha; consumption of working solution = 300.0 L/ha;N100P110K130 + preplanting treatment of tubers, with consumption of the bacterial suspension (1.0 L/t for consumption of working solution and 10.0 L/t for spraying of the plants in the vegetative growth stage first and in the flowering stage second); consumption of bacterial suspension = 3.0 L/ha; consumption of working solution = 300.0 L/ha.

The area of the experimental plot was 100 m^2^, and the area of the accounting plot was 25 m^2^. The experiment was performed in four replications. The planting scheme was 75 × 30 cm, which means the planting density was 44,000 pieces of tubers per 1 hectare. By the period of harvesting, the density of standing plants averaged 43,500 pieces per hectare.

The weight of the planting tubers was 70–80 g. The soil of the experimental field was sod-podzolic sandy loam. The predecessor was annual herbs. Organic fertilizers were not applied for the potatoes, and mineral fertilizers were applied when forming combs at the rate of N100P110K130 kg per ha. Plant care was carried out with row-to-row treatments (two pre-emergence and one post-emergence), spraying of the plants with Titus herbicide (Dupont Science and Technology, Wilmington, DE, USA) (0.03 kg/ha) with Trend (Dupont Science and Technology, Wilmington, DE, USA) (0.2 kg/ha) and Actara (Syngenta, Basel, Switzerland) insecticide (0.06 kg/ha) using an OH-600 sprayer (LLC Zarya, Chelyabinsk city, Russia) with a working fluid flow rate of 300 l/ha. Harvesting was performed with a KTN-2B potato digger (OJSC Lidselmash, Lida, Republic of Belarus), with the collection of tubers being carried out manually before weighing them. The agrochemical characteristics of the soil from the test field were as follows: the content of humus according to Tyurin was 1.70%, the reaction of the medium was pH_KCl_ 4.95, the content of mobile phosphorus was, on average, 342 mg/kg of soil, the exchangeable potassium was 64 mg/kg of soil, and the mineral nitrogen concentration was 24.5 mg/kg of soil (Appendix A).

Field tests with white cabbage were carried out on cv. Amager 611, selected by the All-Russian Research Institute of Vegetable Breeding and Seed Production (Moscow region, Russia). The aim of the field tests was to determine which amount of the TS3 bacterial suspension (1.5 × 10^8^ cfu/mL) had the most pronounced effect on plant yield. The experimental design was as follows:Control = N120P120K180;N120P120K180 + spraying of plants before planting seedlings (consumption of bacterial suspension = 1.0 L/ha, with spraying of plants first 15 days after planting seedlings and second in the phase of forming the head); consumption of bacterial suspension = 1.5 L/ha; consumption of working solution = 300 L/ha;N120P120K180 + spraying of plants before planting seedlings (consumption of bacterial suspension = 1.0 L/ha, with spraying of plants first 15 days after planting seedlings and second in the phase of forming the head); consumption of bacterial suspension = 3.0 L/ha; consumption of working solution = 300 L/ha;N120P120K180 + spraying of plants before planting seedlings (consumption of bacterial suspension = 1.0 L/ha, with spraying of plants first 15 days after planting seedlings and second in the phase of forming the head); consumption of bacterial suspension = 4.5 L/ha; consumption of working solution = 300 L/ha.

The area of the experimental plot was 20 m^2^, and the area of the accounting plot was 10 m^2^. The experiment was performed in four replications. The soil of the experimental field was a sod-podzolic loamy medium. The agrochemical characteristics of the arable (0–20 cm) soil layer before planting the seedlings were as follows: the content of humus according to Tyurin was 1.62%, the reaction of the medium was pH_KCl_ 6.1, the content of mobile phosphorus was, on average, 472 mg/kg of soil, the exchangeable potassium content was 167 mg/kg of soil, and the mineral nitrogen content was 9 mg/kg of soil (Appendix A). The predecessor of cabbage was vegetable peas. In the spring, harrowing with a BZSS 1.0 harrow (LLC Penzagroremmash, Penza, Russia) was carried out to prevent the soil from drying out. Before planting the cabbage seedlings, two continuous forms of tillage were carried out to a depth of 4–6 and 6–8 cm to control weeds. The sowing of seeds for the seedlings was carried out in cassettes. Before planting the cabbage seedlings, ridges were formed using the AF 140 Maschio Gaspardo ridge former (Maschio Gaspardo S.p.A., Campodarsego, Italy). The cabbage seedlings were planted manually with a planting density of 35,000 plants/ha. During the growing season, mechanized row-to-row processing and manual weeding were carried out. To combat leaf-eating pests, fourfold treatment with BI-58 insecticides (BASF, Ludwigshafen, Germany) (400g/L) and Decis-profi (Bayer AG Crop Science Division, Monheim am Rhein, Germany) (0.03 L/ha) was carried out. Foliar spraying of plants was carried out by spraying the plants with a SOLO 444 sprayer (SOLO Kleinmotoren GmbH, Sindelfingen, Germany) with a consumption rate of 300 L/ha. Harvesting was performed manually, and the collected cabbages were weighted on the scales. Briefly, all cabbage heads from each plot were cut with a knife and weighted on a big scale in the field.

### 4.9. Genome Analysis

Genome analysis was carried out as described in [39]. Genome annotation was performed using the NCBI Prokaryotic Genome Annotation Pipeline (PGAP-6.1) [90].

### 4.10. Statistical Analysis

Statistical analysis was performed in the R environment [91] using Rstudio. The *p* values were obtained using one-way ANOVA followed by Tukey’s post hoc test. Significance was computed at *p* < 0.05.

## 5. Conclusions

We isolated the endophyte *Bacillus safensis* TS3 from the stems of tomatoes grown in salty soils, which was capable of producing proteases, lipases, beta-glucanases, and cellulases as well as a range of phytohormones such as IAA, ABA, and tZ. We also demonstrated that the strain TS3 was able effectively colonize the endosphere of plants and was a good root colonizer. These traits are quite important for delivery to the host plants’ produced phytohormones. We demonstrated that the strain TS3 was able to grow on the nutrient medium with 1–15% NaCl and stimulate the growth of winter wheat plants under salinization conditions. The compensatory effect of the strain TS3 for treated plants ranged from 35.5% to 44.5% for the shoot length and varied from 27.9% to 29.1% for the root length. A plant growth-promoting effect of strain TS3 of *B. safensis* was confirmed in the field tests with potato and cabbage plants, which resulted in a yield increase of 8.4–29.4% or 3.4–14.5 t/ha. In the pot experiments with radish and oat plants in salinization, we demonstrated that the strain TS3 contributed to the partial removal of the negative effect of salinization. The compensatory effect of strain TS3 on radish plants during salinization was 46.7%, and for oats, it was 108%. We suppose that such a pronounced effect on the plants grown and the salt stress is connected with its ability to produce phytohormones such as IAA, ABA, and tZ. Genome analysis of strain TS3 of *B. safensis* showed the presence of the necessary genes for the synthesis of proline *pro*A and *pro*B, two different glycine betain transport systems, and a number of antioxidant enzymes, such as superoxide dismutase, catalase, and glutathione reductase. However, the production of these compounds by strain TS3 of *B. safensis* will still need to be demonstrated. Strain TS3 of *B. safensis* can be considered a promising candidate for developing biofertilizer to alleviate salt stress and increase plant yield.

## Figures and Tables

**Figure 1 plants-13-00075-f001:**
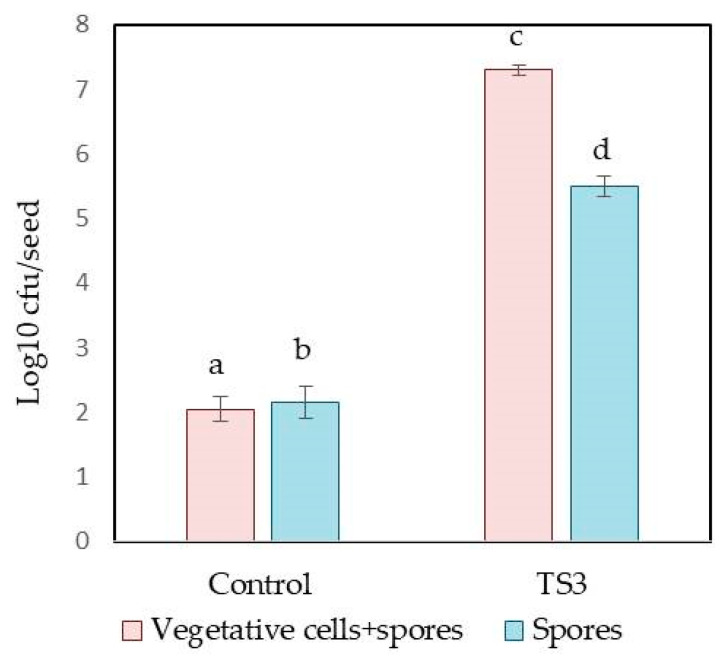
Comparison of cultivated microbiome in untreated and treated seeds of rapeseed plants. Control = seeds incubated with sterile distilled water. TS3 = seeds immersed in the cell suspension of TS3. Different lowercase letters represent values with statistically significant differences (*p* value < 0.05), as determined by Tukey’s test. Bars represent the mean ± SD of four replications.

**Figure 2 plants-13-00075-f002:**
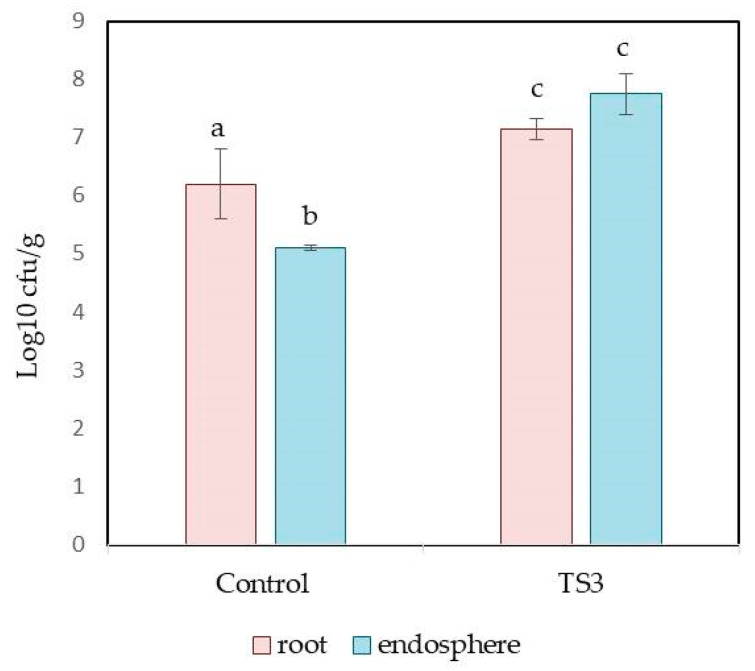
Comparison of cultivated root and endosphere microbiome of two-week-old rapeseed seedlings in the control and when treated with the TS3 variants. Control = seeds incubated with sterile distilled water. TS3 = seeds immersed in the cell suspension of TS3. Bars represent the mean ± SD of eight replications. Different lowercase letters represent values with statistically significant differences (*p* value < 0.05), as determined by Tukey’s test.

**Figure 3 plants-13-00075-f003:**
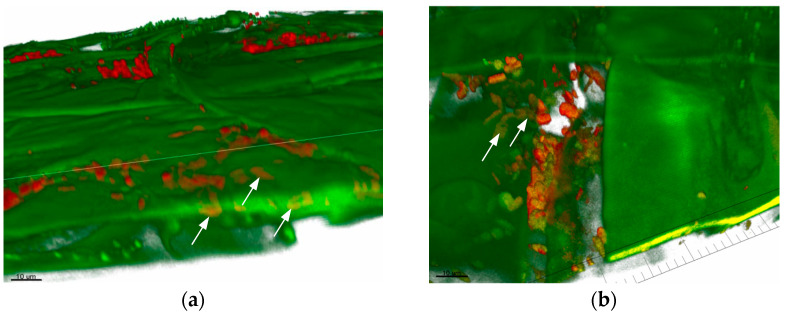
Colonization pattern of *Bacillus safensis* TS3, with TS3 showing endophytic colonization (arrows). IMARIS-edited CLSM picture of *B. safensis* TS3-treated oilseed rape and old root parts, showing rendered root parts (green) and firmicutes (red on the surface and yellow inside the tissue). (**a**) TS3 endophytic colonization (yellow inside the tissue). (**b**) TS3 densely packed colonies.

**Figure 4 plants-13-00075-f004:**
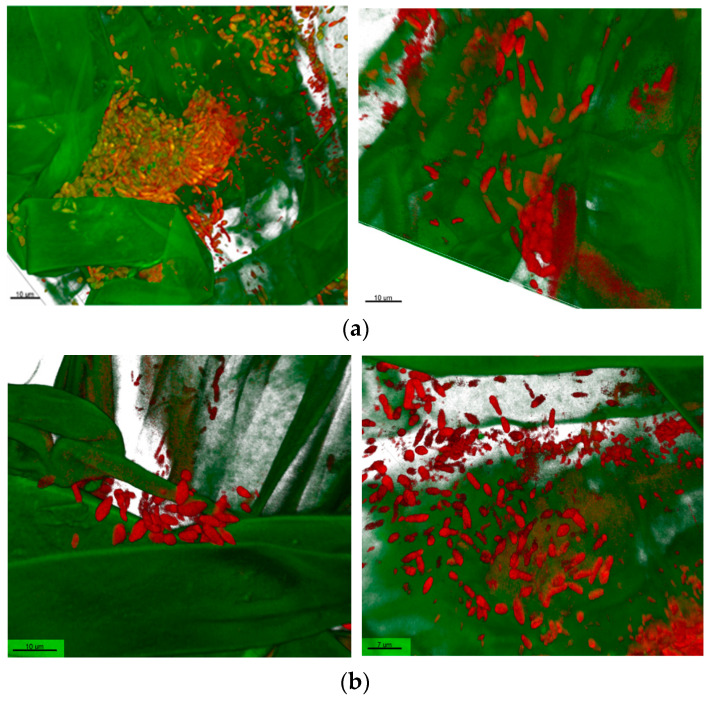
Colonization pattern of *Bacillus safensis* TS3. IMARIS-edited CLSM picture of *Bacillus safensis* TS3-treated oilseed rape and old root parts, showing rendered root parts (green) and firmicutes (red). (**a**) *Bacillus safensis* TS3 dense colonies. (**b**) *Bacillus safensis* TS3 loose colonies.

**Figure 5 plants-13-00075-f005:**
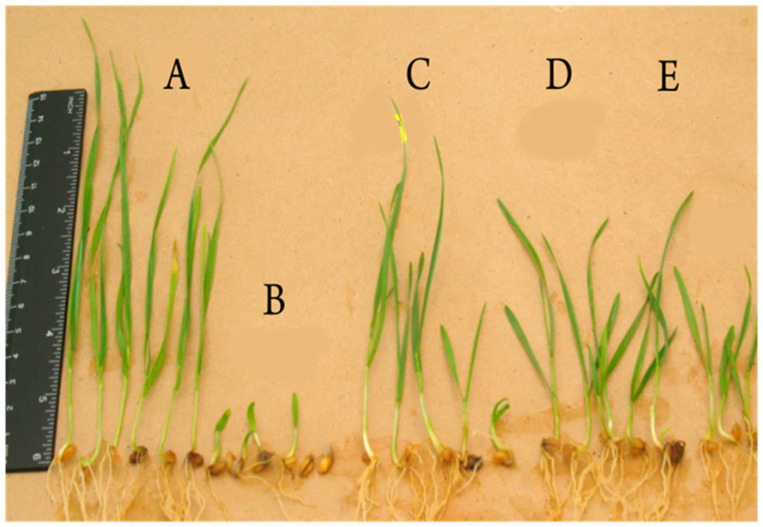
The effect of inoculation with strain TS3 under the action of 1% NaCl. A = negative control; B = positive control; C = 10^7^ CFU/mL; D = (10^6^ CFU/mL); E = (10^5^ CFU/mL). Variant without NaCl was considered the negative control, while the variant with a content of 1% NaCl was considered the positive control.

**Figure 6 plants-13-00075-f006:**
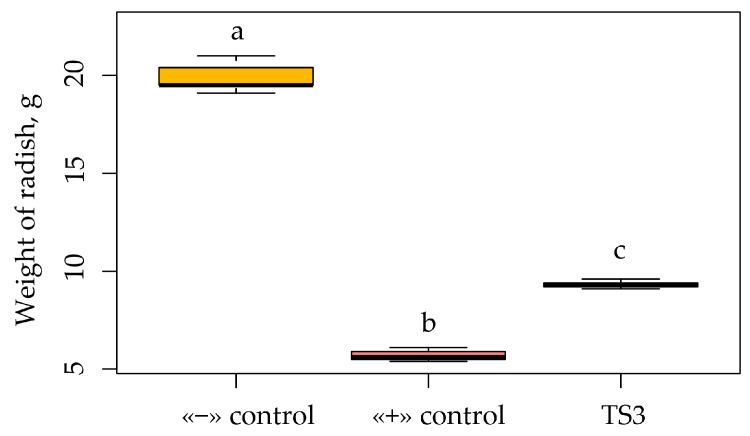
Compensatory capabilities of strain TS3 on radish cv. Duro plants under salinization conditions. Variant without NaCl was considered the negative control (“−” control), while variant with content of 1% NaCl was considered the positive control (“+” control). Boxplots were generated using boxplot package in R. Different lowercase letters represent values with statistically significant differences (*p* value < 0.05), as determined by Tukey’s test.

**Figure 7 plants-13-00075-f007:**
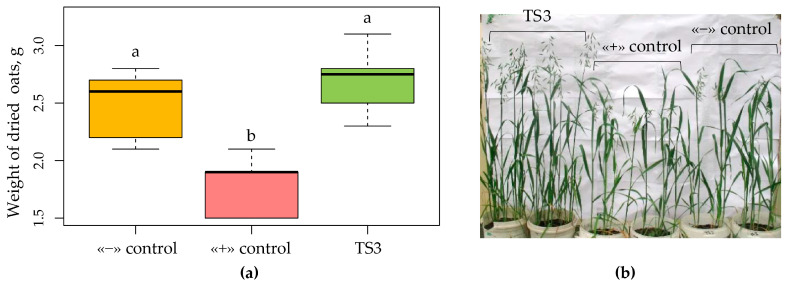
The effect of inoculation of oats with strain TS3 on plant development under salinization conditions. (**a**) Compensatory capabilities of strain TS3 on oat cv. Borrus under salinization conditions. Boxplots were generated using boxplot package in R. Different lowercase letters represent values with statistically significant differences (*p* value < 0.05), as determined by Tukey’s test. (**b**) On the far left is TS3 inoculation, in the center is the positive control (salinization), and on the far right is the negative control (without salinization).

**Figure 8 plants-13-00075-f008:**
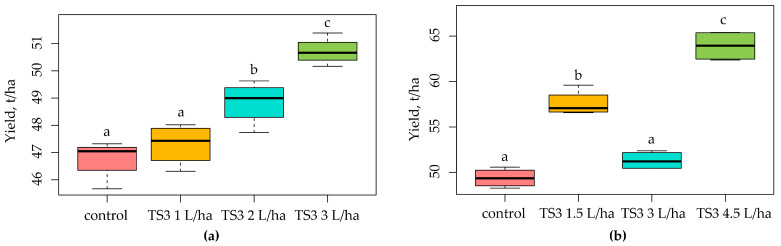
Yield of potato cv. Sante (**a**) and cabbage cv. Amager (**b**) in field tests with *Bacillus safensis* TS3. Boxplots were generated using boxplot package in R. Different lowercase letters represent values with statistically significant differences (*p* value < 0.05), as determined by Tukey’s test.

**Figure 9 plants-13-00075-f009:**
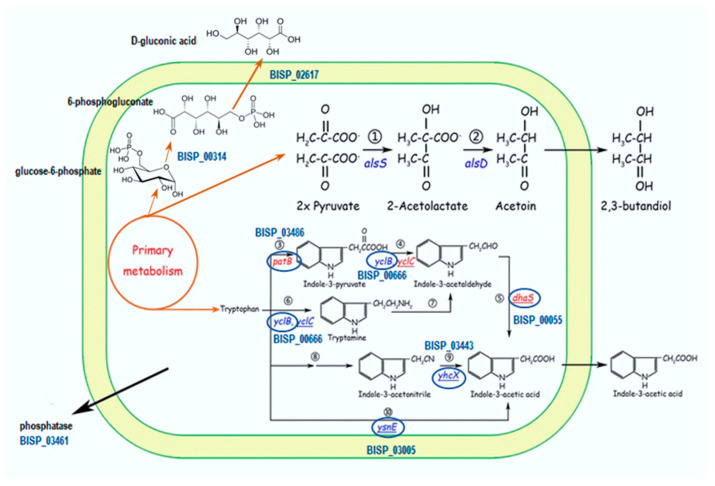
Pathways involved in the synthesis of plant growth-promoting metabolites in TS3. Enzymes marked with circles were predicted in the TS3 genome, suggesting three putative routes for IAA biosynthesis in this strain. TS3 also has all the genes involved in 2,3 butandiol and D-gluconic acid synthesis, as well as a gene encoding a secreted phosphatase. Figure adapted from [41].

**Figure 10 plants-13-00075-f010:**
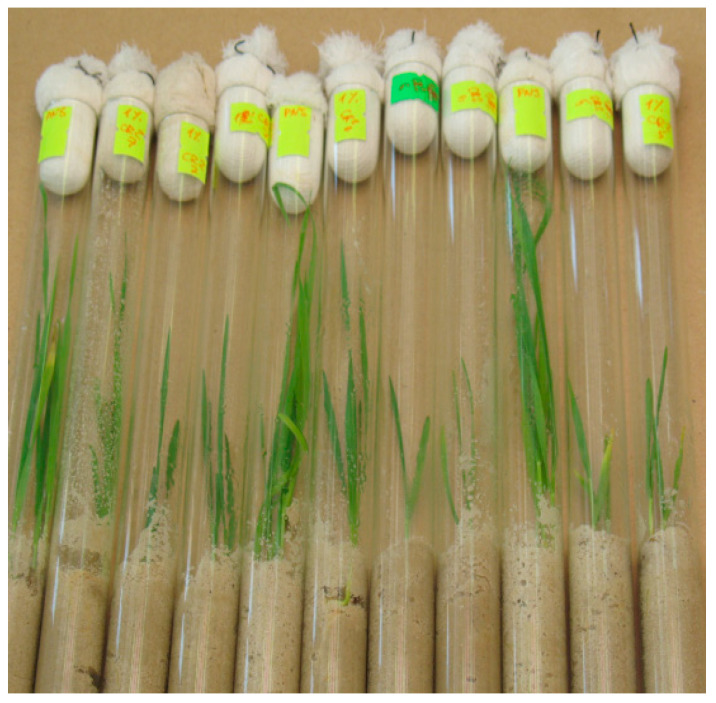
Model gnotobiotic systems with winter wheat plants to study the growth-stimulating effect of bacteria.

**Table 1 plants-13-00075-t001:** Capability of the strain TS3 of *Bacillus safensis* to produce enzymes.

Proteases (Halo Zone) (mm)	Lipases (Halo Zone) (mm)	Beta-Glucanases (Halo Zone) (mm)	Cellulases (Halo Zone) (mm)
12.7 ± 1.1	7.1 ± 0.9	2.5 ± 0.03	17.3 ± 1.1

**Table 2 plants-13-00075-t002:** The concentrations of IAA, ABA, and tZ in the bacterial suspension of *Bacillus safensis* TS3.

IAA (µg/L)	ABA (µg/L)	tZ (µg/L)
366.1 ± 12.5	0.646 ± 0.02	0.822 ± 0.04

**Table 3 plants-13-00075-t003:** The effect of different concentrations of NaCl on the cell number of strain TS3 for *B. safensis* (×10^6^ cfu/mL).

1% NaCl	5% NaCl	10% NaCl	15% NaCl	20% NaCl
109.2 ± 8.7	33.5 ± 2.7	12.6 ± 1.3	0.7 ± 0.03	0

**Table 4 plants-13-00075-t004:** The effect of various NaCl concentrations on the growth of radish, oats, and winter wheat in gnotobiotic systems [40].

NaCl Concentration	Average Length of Seedling (mm)	Average Length of Root (mm)
**Radish**		
0	30.1 ± 2.2 ^a^	61.0 ± 4.1 ^a^
0.5	23.3 ± 1.3 ^b^	51.5 ± 3.5 ^b^
1.0	18.7 ± 1.0 ^c^	38.7 ± 2.5 ^c^
1.5	12.5 ± 0.9 ^d^	11.3 ± 1.5 ^d^
2.0	4.5 ± 0.5 ^e^	6.1 ± 0.5 ^e^
2.5	-	-
5.0	-	-
**Oats**		
0	83.5 ± 5.5 ^a^	106.1 ± 8.3 ^a^
0.5	52.5 ± 3.7 ^b^	87.3 ± 6.1 ^b^
1.0	29.1 ± 2.3 ^c^	63.5 ± 4.7 ^c^
1.5	16.7 ± 0.9 ^d^	35.1 ± 1.7 ^d^
2.0	9.3 ± 0.5 ^e^	23.7 ± 1.1 ^e^
2.5	3.3 ± 0.3 ^f^	5.5 ± 0.3 ^f^
5.0	-	-
**Winter Wheat**		
0	43.3 ± 2.9 ^a^	89.1 ± 6.7 ^a^
0.5	26.7 ± 1.5 ^b^	48.3 ± 3.1 ^b^
1.0	19.5 ± 1.1 ^c^	37.1 ± 2.0 ^c^
1.5	14.7 ± 0.7 ^d^	23.1 ± 1.3 ^d^
2.0	5.6 ± 0.3 ^e^	10.1 ± 0.5 ^e^
2.5	-	
5.0	-	

Different lowercase letters represent values with statistically significant differences (*p* value < 0.05), as determined by Tukey’s test.

**Table 5 plants-13-00075-t005:** Effect of inoculation of winter wheat cv. Batko with various concentrations of the strain TS3 in salinated sand.

Variant	Shoot Length (mm)	Root Length (mm)
Negative control without NaCl	166 ± 10.1 ^a^	79 ± 5.3 ^a^
Positive control with NaCl	26 ± 1.9 ^d^	29 ± 1.5 ^b^
NaCl + TS3, 10^5^ CFU/mL	85 ± 7.6 ^c^	51 ± 4.7 ^c^
NaCl + TS3, 10^6^ CFU/mL	90 ± 7.9 ^c^	52 ± 4.5 ^c^
NaCl + TS3, 10^7^ CFU/mL	100 ± 8.3 ^b^	52 ± 4.3 ^c^

Different lowercase letters represent values with statistically significant differences (*p* value < 0.05), as determined by Tukey’s test.

## Data Availability

The data presented in this study are available upon request from the corresponding author.

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
