# Peer review of "Phytohormone Production by the Endophyte Bacillus safensis TS3 Increases Plant Yield and Alleviates Salt Stress"

_plants, 2023, doi:10.3390/plants13010075_

Round 1
Reviewer 1 Report
Comments and Suggestions for Authors
In my opinion, the study of the effectiveness of Bacillus safensis (SF3 isolated strain) in improving tolerance to salt stress in various crops is interesting. Although the work seems to be well done, the writing of the article should be improved to make it easier to read. Regarding what I consider most notable, in terms of material and methods, the information should be presented sequentially and more clearly. All agronomic details should be included, some protocols briefly described, and the statistics section better explained. Different tests have been carried out on different crops, I do not understand why a single crop or several have not been taken, and all the tests have been carried out to verify that they are really fulfilled in all of them. At the level of results, there are results that are not included but are talked about, and results that are not shown, or in which statistical analysis is not applied. The legends in many cases should be improved. At the discussion level, it should be justified what improvement SF3 provides compared to other strains, and highlight what is novel in this study, since the positive effect of B. safensis is already known. The format should be reviewed and the English reviewed by a native person. Additional comments are included in the attached document.

Comments on the Quality of English LanguageEnglish should be reviewed by a native English person.
Reviewer 2 Report
Comments and Suggestions for Authors
I suggest reviewing the introduction since there is similarity with other articles. The discussion should also be improved; in its current state, it is very descriptive, and does not incorporate an exhaustive analysis of the results shown. A good example of this is the last sentence of the introduction
Thus, it can be concluded that the production of phytohormones such as IAA, ABA and tZ by the strain TS3 B. safensis contributed to alleviation of salt stress and increase of plants yield. At the same time, genome analysis of the strain TS3 B. safensis showed the presence of the necessary genes for the synthesis of proline proA and proB, two different glycine betain transport systems, and a number of antioxidant enzymes, such as superoxide dismutase, catalase and glutathione reductase. Therefore, the production of these compounds by the strain TS3 B. safensis and protective ability will still need to be demonstrated.
Round 2
Reviewer 1 Report
Comments and Suggestions for Authors
First of all, I would like to thank the authors for answering my questions. I was able to verify that most of my suggestions have been implemented in the manuscript. I think the manuscript has been improved enough for publication, although I think the following should be completed:
- The numerical results of enzyme production and the effect of different concentrations of NaCl relative to TS3 are not given in Tables 1 and 3.
- Table 4 does not show the statistics related to possible significant differences.
- Table 5 does not indicate the number of replicates per treatment, nor has the statistical treatment of the data been carried out.
- Figure 5 should show the effect on roots.
Comments on the Quality of English LanguageIn my opinion, minor editing of English language is required.
